# Exhausted T cell signature predicts immunotherapy response in ER-positive breast cancer

Manuela Terranova-Barberio[1], Nela Pawlowska[1], Mallika Dhawan[1], Mark Moasser [1], Amy J. Chien[1], Michelle E. Melisko[1], Hope Rugo[1], Roshun Rahimi[1], Travis Deal[1], Adil Daud [1], Michael D. Rosenblum[2], Scott Thomas[1] & Pamela N. Munster [1]✉

Responses to immunotherapy are uncommon in estrogen receptor (ER)-positive breast cancer and to date, lack predictive markers. This randomized phase II study defines safety and response rate of epigenetic priming in ER-positive breast cancer patients treated with checkpoint inhibitors as primary endpoints. Secondary and exploratory endpoints included PD-L1 modulation and T-cell immune-signatures. 34 patients received vorinostat, tamoxifen and pembrolizumab with no excessive toxicity after progression on a median of five prior metastatic regimens. Objective response was 4% and clinical benefit rate (CR + PR + SD > 6 m) was 19%. T-cell exhaustion (CD8$^+$ PD-1$^+$/CTLA-4$^+$) and treatment-induced depletion of regulatory T-cells (CD4$^+$ Foxp3$^+$/CTLA-4$^+$) was seen in tumor or blood in 5/5 patients with clinical benefit, but only in one non-responder. Tumor lymphocyte infiltration was 0.17%. Only two non-responders had PD-L1 expression >1%. This data defines a novel immune signature in PD-L1-negative ER-positive breast cancer patients who are more likely to benefit from immune-checkpoint and histone deacetylase inhibition (NCT02395627).

[1] Division of Hematology and Oncology, University of California, San Francisco, CA, USA. [2] Department of Dermatology, University of California, San Francisco, CA, USA. ✉email: pamela.munster@ucsf.edu

B reast cancer remains the most commonly diagnosed non-skin cancer for women in the US with two thirds of all breast cancers expressing hormone receptors[1]. Cell-based immunotherapy and checkpoint inhibitors have revolutionized therapy for many cancers. However, for patients with hormone receptor positive breast cancer, immune modulation has had only minimal efficacy, and meaningful responses appear to be limited to triple negative breast cancer (TNBC). In these tumors, single agent checkpoint inhibition showed a response rate ranging from 5–24%, whereas an overall response rate of 12% or less was found in estrogen receptor (ER-)positive breast cancer patients[2–5]. In hormone responsive metastatic patients, no benefit was obtained by the addition of pembrolizumab to eribulin mesylate chemotherapy[6]. Higher PD-L1 expression in breast cancer was associated with better response and survival in several reports[3,7–10]. The response rate to PD-1 checkpoint inhibitors in patients with low or absent PD-L1 expression was less than 5% and responses were rarely seen in patients who had previously been exposed to multiple lines of therapy for their metastatic disease[3,11]. The recent approval of the PD-L1 checkpoint inhibitor, atezolizumab, in combination with chemotherapy in PD-L1-positive TNBC further confirms the findings of smaller studies. This large randomized study showed an improvement, albeit modest in tumors with high PD-L1 expression, but in tumors not expressing PD-L1, no benefits were seen with atezolizumab, even when combined with chemotherapy[5,12].

Other factors implicated in the lack of response or resistance to immunotherapy include absence of tumor infiltrating lymphocytes (TILs), lack of tumor antigenicity, dysregulation of WNT-β-catenin pathway[13,14], PTEN loss[15], p53 loss[16] and deletional mutations in the JAK1/2–STAT[17] signaling pathway. Moreover, the gut microbiome, the presence of female gender and liver metastases have also been implicated in lack of response, as well as high baseline immune markers such as IL6 and C-reactive protein[18–21]. In addition to many of these confounders, ER-positive tumors have less abundant TILs and PD-L1 expression than TNBC or HER2 positive breast cancers[9,10,22]. However, recent data suggest that while high TILs infiltration is associated with better outcome overall, only immune infiltrates expressing PD-1 and PD-L1 appear to be relevant in the response to immune checkpoint inhibitors[12]. Both high PD-L1 expression and TILs are particularly scarce in women with ER-positive tumors, which prompted pretreatment with an epigenetic immune modulator in our study.

Histone deacetylase inhibitors (HDACi) such as vorinostat, are epigenetic modifiers that have been shown to reverse hormone therapy resistance, resulting in prolonged anti-tumor responses in patients. In particular, vorinostat has shown to restore hormone therapy sensitivity in an ER-positive breast cancer preclinical model, by inhibiting autophagy and redirecting cells into apoptosis[23]. The integration of HDACi into hormonal therapy was tested in two clinical trials performed by our group. In the first trial, vorinostat was combined with tamoxifen in patients with heavily pre-treated, ER-positive advanced breast cancer[24], obtaining an objective response rate (ORR) of 19% and clinical benefits rate (ORR and stable disease >24 weeks) of 40%. In a second trial (ENCORE 301), entinostat was combined with exemestane, improving the median overall survival (OS) from 19.8 months, in the exemestane only arm, to 28.1 months in the combination arm[25], with a large phase III trial having recently completed accrual.

In addition to their effects on ER signaling, preclinical studies suggested that HDACi reduce regulatory T-cells (Tregs), induce PD-L1 expression on tumor cells and change the composition of TILs, specifically inducing CD8+ T-cells in vitro and in vivo in breast cancer models[22,26,27].

Thus, we leveraged the ability to inhibit ER signaling and modulate tumor immunity, and conducted a randomized phase II clinical trial combining vorinostat with the selective ER modulator tamoxifen and the immune checkpoint inhibitor pembrolizumab (NCT02395627), given either on day 1 of cycle 1 or on day 1 of cycle 2. The aim was to determine whether co-administration of a HDACi, can convert immune silent ER-positive breast cancers into an immune responsive phenotype and define any signatures of response for this combination.

## Results

**Study objectives**. The primary endpoints of this study were to define the optimal approach for epigenetic priming in breast cancer patients on the basis of overall response and to define safety and tolerability. The secondary endpoints were to assess duration of response and the impact of vorinostat on PD-L1 expression in ER-positive tumors. Finally, the exploratory endpoints include post-treatment assessment of TILs population, inflammatory T-cell signature and PBMCs histone acetylation as a pharmacodynamic response marker for HDACi treatment.

**Baseline characteristics and patient disposition**. Thirty-four heavily pretreated ER-positive breast cancer patients [median age 56 years (32–81)] were enrolled and received at least one dose of therapy. The median number of prior lines of systemic treatment in this metastatic setting was 5 (range 2–13), with 85% of patients having received at least 3 lines of therapy, including hormonal therapy. Only 2 patients had tumors with more than 1% of PD-L1 expression (Supplementary Fig. 1). Due to limited ORR and insufficient clinical benefit in an unselected patient population, the accrual was halted prior to completion of target enrollment, in accordance with the study's protocol. Baseline characteristics for the patients are shown in Table 1.

**Efficacy and toxicity analyses**. Thirty-four patients were treated on this trial prior to stopping the study. Six of the 34 patients never received pembrolizumab due to progression prior to scheduled infusion (Fig. 1). One patient was taken off study at week 3, without evidence of progression, due to immune-related grade 3 hepatitis (Fig. 1). A complete response (CR) was seen in one of the 27 (3.7%) evaluable patients who received all three agents with a clinical benefits rate [CBR: CR, partial response (PR) and stable disease (SD) > 6 months] of 5/27 patients (18.5%) and an objective response rate of 1/27 (3.7%). None of the patients with clinical benefit had discernible PD-L1 staining in their tumor. The patient with the CR had her treatment stopped after 10 months due to a disabling stroke. Upon a follow up visit at 21.7 months, she had not progressed and was lost to follow up afterwards. The median duration of treatment for all patients was 3.4 months (range 1–22+ months). These findings suggested that epigenetic modulation or priming did not benefit this group of heavily pretreated, ER-positive patients. The two patients on study with detectable PD-L1 expression experienced no benefit from therapy.

Overall the combination treatment was well tolerated and the addition of the HDACi did not worsen pembrolizumab or tamoxifen adverse events. Grade 3/4 toxicities included elevated transaminitis (9%, including immune hepatitis requiring treatment discontinuation), fatigue (6%), hyponatremia (6%), thrombocytopenia (6%) and anorexia (3%), as well as a patient with a disabling stroke of unclear relatedness, requiring treatment discontinuation. Grade 2 immune-related toxicities included pneumonitis, hypothyroidism, colitis, fatigue and thrombocytopenia (Table 2).

**Table 1 Baseline patient demographic and clinical characteristics.**

| Baseline characteristic ($N = 34$) | |
|---|---|
| **Median age, n (range)** | 59 (32–81) |
| **Gender, n (%)** | |
| Male | 1 (3) |
| Female | 33 (97) |
| **Race/Ethnicity, n (%)** | |
| Caucasian | 22 (64) |
| African-American | 4 (12) |
| Asian | 4 (12) |
| Hispanic | 4 (12) |
| **ECOG performance status, n (%)** | |
| 0 | 4 (12) |
| 1 | 30 (88) |
| **Patients disposition** | |
| Patients evaluable for pembrolizumab response, n (%) | 27 (79) |
| Reasons for study discontinuation, n (%) | |
| Disease progression | 27 (79) |
| Withdrawal related to pembrolizumab toxicity | 1 (3) |
| Withdrawal of consent before pembrolizumab | 2 (6) |
| Withdrawal for progression before pembrolizumab | 4 (12) |
| Prior therapy in metastatic setting, median (range) | |
| Total regimens | 5 (2–13) |
| Chemotherapy | 2 (0–8) |
| Hormonal therapy | 3 (0–5) |
| Biologic therapy | 2 (0–4) |
| Patients receiving prior tamoxifen, n (%) | 9 (26) |
| TIL and PD-L1 expression | |
| CD3+, median (range) | 0.15 (0.005–1.65) |
| PD-L1, n (%) | 2 (6) |

Data include a total of 34 patients.
*ECOG* Eastern Cooperative Oncology Grade.

**Correlative analysis and immune response modulation**. The role of TILs and partially exhausted PD-1+/CTLA4+ CD8+ T-cells[28] and their epigenetic modulation were part of preplanned correlative endpoints in blood and tumor samples for this study, as well as comprehensive T-cell immune-profiling. The proportion of responders (CR, PR and SD > 6 months) and duration of response was markedly improved in patients with an increased number of exhausted cytotoxic CD8+ T lymphocytes (CTLs defined as CTLA4+/PD-1+ CD8+ T-cells) in either blood or tumor, with longer median progression-free survival (PFS) (8.6 versus 2.8 months, HR 0.32, $p = 0.0098$) (Fig. 2a, b and Supplementary Fig. 2). Similarly, longer median OS was observed, but did not reach statistical significance (29.50 versus 15.23 months, HR = 0.51 and $p = 0.12$). The percentage of exhausted CTLs was significantly higher in patients with response versus non-responders (41.03 vs 3.93%, $p < 0.0001$) (Fig. 2c, d). Clinical benefit (CR, PR and SD > 6 months) and PFS strongly correlated with a higher percentage of partially exhausted CTLs in tumor samples (Fig. 2e). This correlation remained significant (with exhausted CTLs in responders and non-responders, 12.97% vs 5.47% respectively, $p = 0.032$) in peripheral blood samples (Supplementary Figs. 3 and 2f). Pearson correlation analysis of the relationship between peripheral blood and tumors was performed in both non-responders and responders. For the non-responders, a significant $p$-value of 0.0178 was found (Supplementary Fig. 4). None of the evaluable non-responders had a CTLA4+/PD-1+ exhausted CTL signature in tumor samples, rendering the positive and negative predictive value of the test at 100%. The negative predictive value for blood was 91%. On the other hand, correlative analysis could not be performed in the

responders, due to lack of tumor tissue that was available for only 3 of the responders. The percentage of exhausted CTLs was not affected by the treatment in either responders or non-responders (Fig. 3a).

Several reports support the correlation of total TILs to immunotherapy response. The overall number of CD3+ cells in tumor samples was very low in all patients, with median number of CD3+ cells relative to tumor cells at baseline (median 0.17% range 0.005–1.6, $N = 22$), which was not affected by treatment (median 0.16%, range 0.03–0.56, $p =$ not significant) (Fig. 3b). Similarly, CD45+ cell infiltration was low (median 0.22%, range 0.006–1.89%) (Fig. 3c).

The overall percentage of CD4+ and CD8+ T-cells was not affected by either pembrolizumab or vorinostat treatment and did not predict response in tumor (Fig. 3d).

Our data did not suggest activation or proliferation of CD8+ T-cells, measured by HLA-DR and Ki67 expression, respectively[28–30], either at baseline or following therapy in tumor or blood (Fig. 3e, f and Supplementary Fig. 5). PD-L1 tumor cell expression was absent in all but 2 patients (both non-responders) at baseline, and unlike observed in TNBC preclinical models[22], no changes in PD-L1 expression were observed in any of the patients (Fig. 2a, Supplementary Fig. 1).

**Epigenetic modulation of CD4+ regulatory T-cells (Tregs) and histone acetylation**. Multiple preclinical studies suggested that epigenetic modulation could impact the efficacy of immune checkpoint inhibitors. HDAC inhibition, and in particular vorinostat, has shown to significantly reduce Treg numbers and activity[22,27]. Previous reports have further linked the increased presence of Tregs in breast cancer with an aggressive phenotype and reduced OS[31–33]. Our preclinical in vivo and in vitro studies showed that HDACi, including vorinostat, potentiated immune checkpoint inhibitors in in vitro models, reduced tumor burden and prolonged the survival of tumor bearing 4T1 mice by decreasing tumor infiltrating Tregs[22]. In this study, responders had more (but not reaching significance) Tregs at baseline (Foxp3+ CD4+: 15.8% vs 11.5%, $p =$ not significant, Fig. 4a).

However, treatment significantly reduced activated Tregs in the tumor microenvironment: Foxp3+CTLA4+ CD4+ Tregs were reduced in all patients, from 11.8% to 2.9%, $p = 0.0067$ (Fig. 4b). This reduction was much more prominent in the patients with clinical benefit, with a decrease from 10.4% to 2.5% ($p = 0.034$) in non-responders compared to a drop from 20.3% to 4.2% ($p = 0.031$) in responders. Our preclinical data in mice suggested that this was due to vorinostat treatment[22]. Neither HLA-DR nor Ki67 expression in Tregs changed with vorinostat treatment, suggesting the treatment directly affected Treg numbers, without modulating their activity, that was comparable in responders and non-responders (Fig. 4c, d).

To further associate a causal role of vorinostat-induced depletion of Tregs and its relevance to immune checkpoint inhibitor response in this study, we correlated sustained histone H3 and lysine acetylation in peripheral blood mononuclear cells (PBMCs) with clinical benefit and time to progression[24,25,34–36]. Using acetylation as a pharmacodynamic measure of HDAC activity, our data further supported that both increased histone H3 and lysine acetylation significantly correlated with time to progression (Fig. 4e, f).

Preclinical in vivo data further suggested a select expansion of CD8+ T-cells in the tumor microenvironment[22], however, this was not seen in patients. CD8+ T-cells were very few in all tumors at baseline, which did not increase with vorinostat treatment at the approved dose. Similarly, treatment did not affect tumor residing or peripheral CD3+, CD4+, CD8+ T-cells

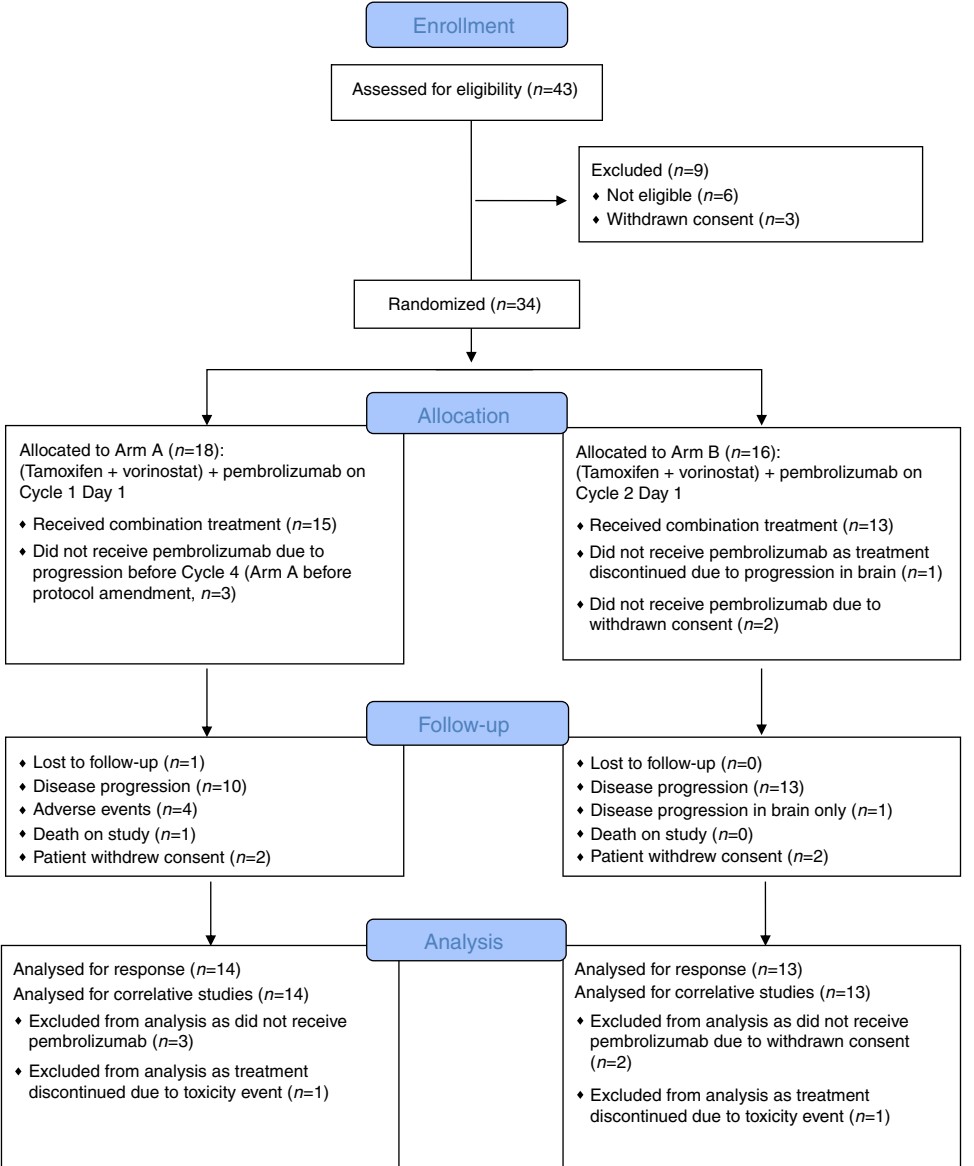

**Fig. 1 CONSORT Flow diagram of the randomized prospective phase II clinical trial.** The consort flow diagram shows the randomization procedures and the description of the two treatment arms of this trial, adding pembrolizumab to tamoxifen and vorinostat either on cycle 1 day 1 or on cycle 2 day 1.

**Table 2 Patients reporting any treatment emergent toxicities.**

| Grade | 1 | 2 | 3 | 4 | 5 |
|---|---|---|---|---|---|
| Arthralgia | 2 (6%) | | | | |
| Anorexia | 8 (24%) | 2 (6%) | 1 (3%) | | |
| Nausea | 8 (24%) | 2 (6%) | | | |
| Diarrhea | 7 (21%) | 3 (9%) | | | |
| Dysgeusia | 4 (12%) | 2 (6%) | | | |
| Fatigue | 6 (18%) | 9 (27%) | 2 (6%) | | |
| Hypothyroidism | 1 (3%) | 1 (3%) | | | |
| Rash | 2 (6%) | | | | |
| AST/ALT | 2 (6%) | 1 (3%) | 3 (9%) | | |
| Thrombocytopenia | 6 (18%) | 4 (12%) | 2 (6%) | | |
| Neutropenia | 2 (6%) | 1 (3%) | | | |
| Hypophosphatemia | 1 (3%) | 1 (3%) | | | |
| Hyponatremia | | | 2 (6%) | | |
| Elevated creatinine | 4 (12%) | 2 (6%) | | | |
| Pneumonitis | | 1 (3%) | | | |
| Stroke | | | | 1 (3%) | |
| Colitis^ | | 1 (3%) | | | |

^>30 days after removal from study.

or their ratio in tumor or blood (Figs. 3d, 5a, b). Our data suggested a strong correlation between response and an increase in exhausted CD8$^+$ T-cells in both tumors and blood. However, peripheral blood Tregs were low (2.95% versus 1.54% responders versus non-responders, respectively, $p=$not significant) and neither their number nor activity were affected by vorinostat on day 12 or after 3 cycles of therapy (Fig. 5c–f).

## Discussion

The important, but limited, response rate to immunotherapy in PD-L1-negative, ER-positive breast cancer lends strong support to further explore the addition of epigenetic modifiers to immune checkpoint inhibitors. In this study, the low response rate is consistent with other immune checkpoint studies enrolling patients with advanced stage disease that received multiple treatment regimens, particularly in patients with PD-L1-negative tumors. Overall, the response seen in our patients, despite the addition of a HDACi, was limited and led to the premature stopping of the trial in unselected patients. Our data, thus, highlight the need for patient selection and we propose an

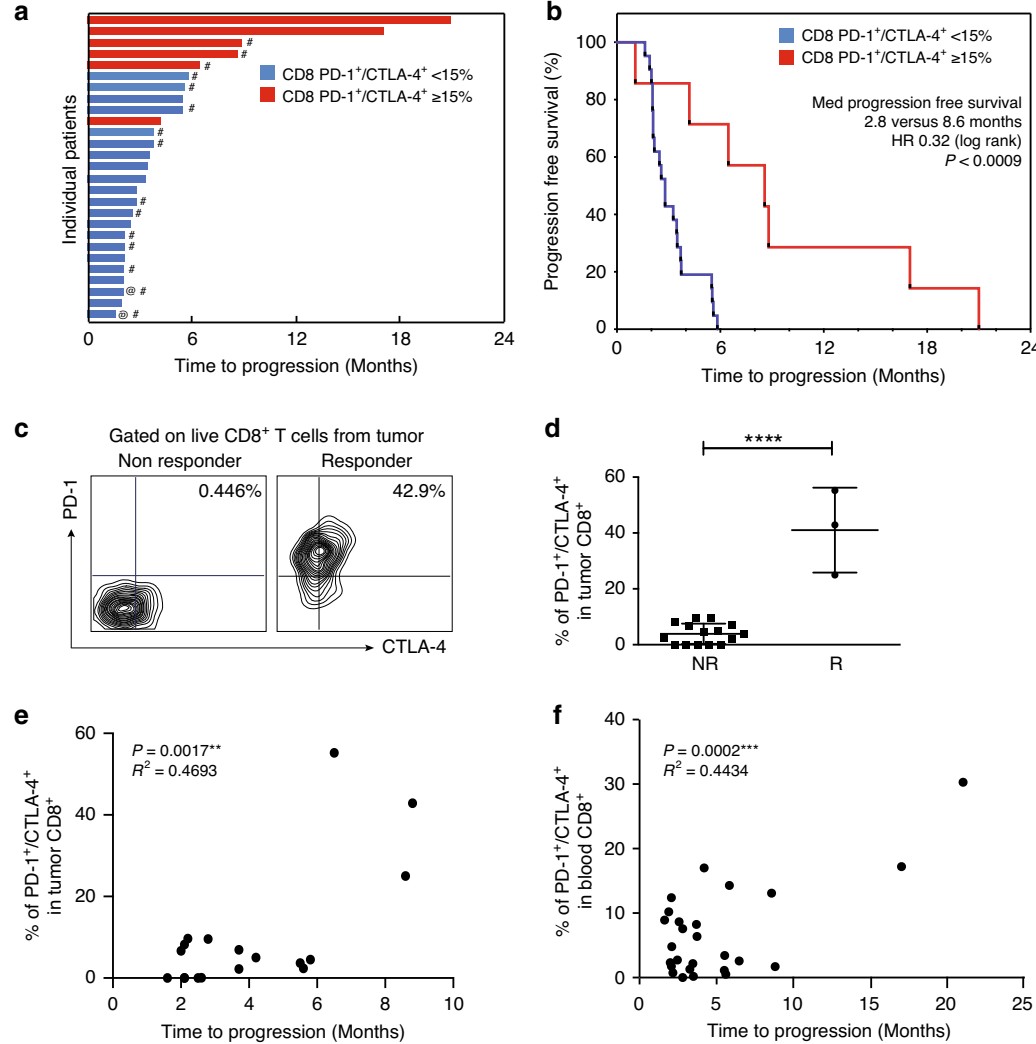

**Fig. 2 Exhausted CD-8 T cells predict response to anti-PD-1 therapy. a** Time to progression in months for the patients' cohort ($n = 27$). **b** Median progression free survival for the patients' cohort. Hashtag (#) indicates tumor specimens available in the analysis; at sign (@) indicates the patient with PD-L1 staining >1%. Patients with PD-1+CTLA-4+ CD8+ T-cells expression [in metastatic tumor sites as hashtag (#), or blood, quantified by flow cytometry] >15% are shown in red, while those with % <15 are shown in blue. **c** Representative flow cytometry plots for PD-1+CTLA-4+ live CD8+ T-cells in tumor in patients who benefit (Responder) or not (Non responder) from therapy. **d** Flow cytometry quantification of PD-1+CTLA-4+ CD8+ T-cells in non-responders (NR) versus responders (R). Data are presented as the mean ± SD ($n = 18$). Statistical significance is indicated by p-values as $*p \leq 0.05$; $**p \leq 0.01$; $****p \leq 0.0001$; NS = not significant and was determined by unpaired two-tailed T test. Correlation of quantification of PD-1+CTLA-4+ CD8+ T-cells from metastatic tumors (**e**) and blood (**f**) with time to progression in months. Two-tailed Pearson correlation test was performed with p-value = 0.0017 and p-value = 0,0002 for E and F respectively.

exhausted CD8+ T-cell immune signature, detectable in both blood and tumor, may predict response to immune checkpoint inhibitors. Moreover, our data suggest that a HDACi-dependent decrease in Tregs contributes to the efficacy observed and warrants further investigation. Indeed, treatment induced a specific decrease in the number of activated Tregs (CD4+/Foxp3+/CTLA4+) in the tumor microenvironment, without modulating Ki67 and HLA-DR activation markers.

There are several limitations to our findings, foremost the limited number of subjects treated. The trial was halted after enrolling 34 of the 87 patients originally planned, due to its limited efficacy in an unselected patient population. Despite the limited number of patients, exhausted T-cell infiltration (CTLA4+/PD-1+ CD8+ T-cells) robustly correlated with response rate. Responder T-cell exhaustion was detectable, despite a very low overall immune cell infiltration, with fewer than 2% of CD45+ cells comprising the tumor microenvironment. While CTL immune

infiltrates are often associated with better outcomes and response to therapy in TNBC, in ER-positive breast cancer, no clear correlation has been found linking immune infiltration with outcomes[37–39].

Further, responders had an increased number of Tregs (FoxP3+/CTLA4+ CD4+ cells) at baseline, which were selectively decreased by the treatment, without affecting the overall CD4+ and CD8+ population or their ratios. Treatment-induced depletion of Foxp3+/CTLA4+ Tregs was only seen in tumor samples, but not in the peripheral blood. A causal, rather than bystander, role of the HDACi in the response to the immune checkpoint inhibitor-HDACi combination is supported by the significant correlation between histone H3 and lysine acetylation in responders and time to progression.

The impact of Tregs in reducing immunotherapy response has been shown across multiple tumor types. In ER-positive breast cancer, Treg tumor infiltration has been linked to worse overall

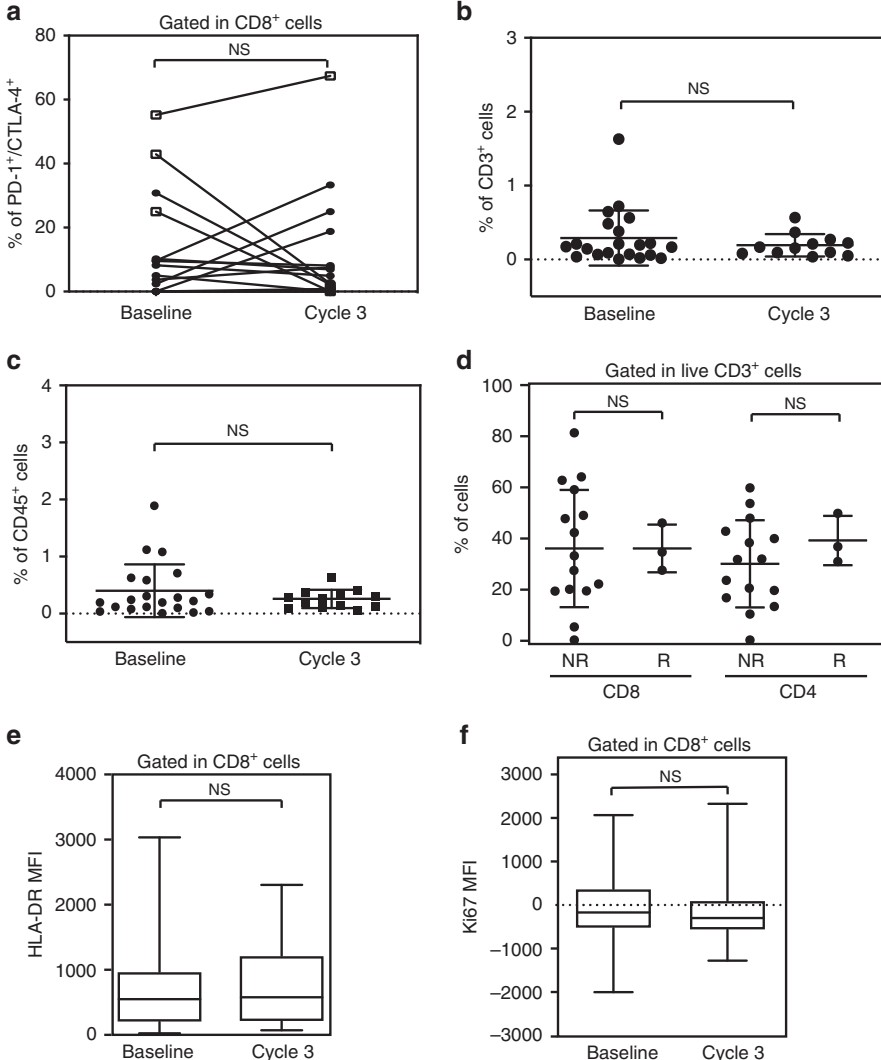

**Fig. 3 No significant change in immune subsets was observed with treatment. a** Line plot for flow cytometry quantification of PD-1+CTLA-4+ live CD8+ T-cells in tumors pre and post therapy on cycle 3. Responders are indicated with an empty square, while non-responders with a full circle ($n = 15$). Flow cytometry quantification of CD3+ (**b**) and CD45+ cells (**c**) at baseline and post treatment. **d** Flow cytometry quantification of CD4+ and CD8+ in the tumor microenvironment in non-responders (NR) versus responders (R). Flow cytometric quantification of HLA-DR (**e**) and Ki67 (**f**) expression in live CD8+ T-cells from metastatic tumors before (Baseline) and after 3 cycles of treatment (Cycle 3) ($n = 14$). The markers represent the minimum, first quartile, median, third quartile, and maximum values, respectively (**e**, **f**). Data are presented as the mean ± SD. Statistical significance is indicated by p-values as *$p \leq 0.05$; **$p \leq 0.01$; ****$p \leq 0.0001$; NS = not significant. This was assessed by two-tailed Wilcoxon text for **a–c** and **e**, **f** and by unpaired T test for D.

outcome[31,33,40–43]. Experimental depletion of Tregs has been associated with improved outcomes and response to immune checkpoint inhibitors[44–46]. Our data show a significant depletion in Tregs with treatment, which was correlated with response in patients with high levels of partially exhausted CTLs. Our data is, further, consistent with reports from other investigators proposing that the increase of Tregs residing in tumors is an epigenetic phenomenon, rather than circulating T-cells infiltrating the tumor[47]. This may also explain why the decrease in Tregs by vorinostat only occurred in tumors and why vorinostat did not affect overall CD4+ or CD8+ cell numbers, their composition or activation measured by HLA-DR and Ki67 expression.

While exhausted CTLs in the tumor immune microenvironment have been shown to correlate with responses in cancers such as melanoma[28], there have been no reports of a predictive value of CTLA4+/PD-1+/CD8+ cells in peripheral blood. The finding of an immune signature in peripheral blood could greatly facilitate patient selection in the clinic with an easily reproducible

and robust flow cytometry assessment. Thus, the next steps will be to evaluate this signature in a prospective trial.

## Methods

**Patient selection.** Pre and postmenopausal women or men with metastatic ER-positive breast cancer, after progression on at least one hormonal therapy in the metastatic setting and any number of prior chemo- or hormonal therapies, including tamoxifen, were eligible for this trial. Measurable disease, defined by RECIST 1.1[48], and pre and post treatment biopsies were required if tumors were accessible and safe for biopsy, together with an ECOG Performance Status ≤ 2 and adequate organ function (absolute neutrophil count ≥ 1.5 ×10^9/L; hemoglobin ≥ 9 g/dL; platelets ≥ 100 ×10^9/L; AST and ALT ≤ 2.5 × Upper Limit Normal (ULN) or ≤ 5.0 × ULN if liver tumor is present; serum total bilirubin ≤ 1.5 × ULN; Serum creatinine ≤ 1.5 × ULN, or 24-h clearance ≥ 60 ml/min). All participants provided written informed consent. The protocol, the proposed informed consent form and all forms of participant information related to the study were reviewed and approved by the UCSF CHR (UCSF Institutional Review Board), the Helen Diller Family Comprehensive Cancer Center Site Committee and by the Protocol Review Committee (PRC).

The trial was initially designed to enroll 87 patients. Using an Optimal Simon's two-stage design[49], the null hypothesis (H_0) that the true response rate is 10% was

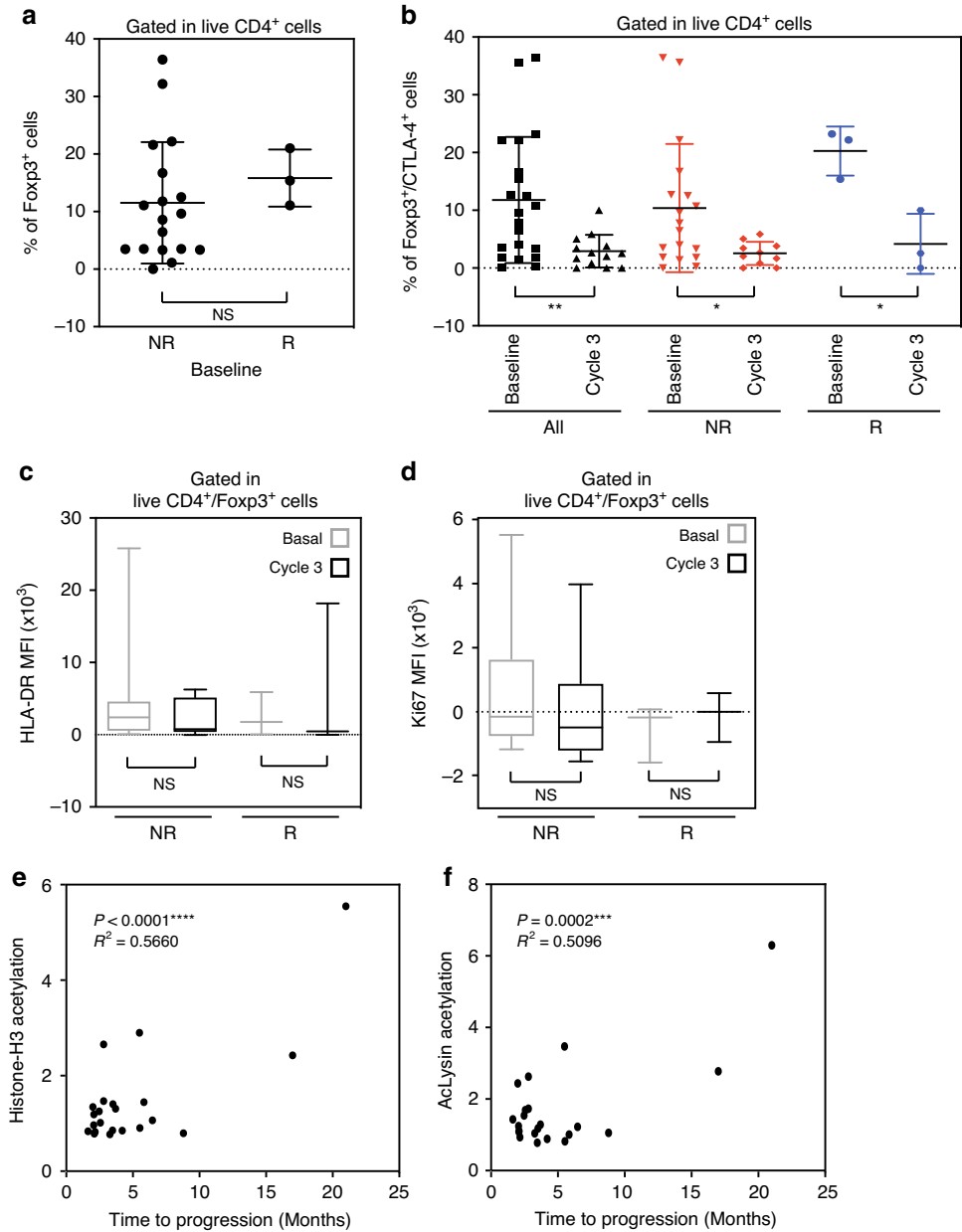

**Fig. 4 Correlative analyses. a** Flow cytometry quantification of tumor Foxp3+ CD4+ Tregs taken from patients before treatments (Baseline) in non-responders (NR) versus responders (R) ($n = 21$). **b** Flow cytometry quantification of tumor Foxp3+ CTLA-4+ CD4+ Tregs in all the patients (black), non-responders (NR, red) and responders (R, blue), before (Baseline) and after 3 cycles of treatment (Cycle 3). **c** Flow cytometric quantification of HLA-DR (**c**) and Ki67 (**d**) expression in Foxp3+ CD4+ T-cells ($n = 12$). The markers represent the 5th percentile, first quartile, median, third quartile, and 95th percentile values, respectively (**c, d**). Two-tailed Pearson correlation between the fold increase in day 10-12 versus day 1 of (**e**) acetyl histone H3 (AcH3) and (**f**) acetyl lysine (AcLys) in lymphocytes and time to progression in months ($n = 22$). Data are presented as the mean ± SD. Statistical significance is indicated by p-values as $p \leq 0.05$; **$p \leq 0.01$; ***$p \leq 0.001$; ****$p \leq 0.0001$; NS = not significant and was determined by two-tailed Mann-Whitney test for **a, b** (All and NR), with two-tailed Wilcoxon test for **c, d**, and by two-tailed paired T test for B (R).

to be tested against a one-sided alternative. The trial protocol stipulated that in the first stage, 10 patients would be accrued, if there were 0 or 1 objective responses at 24 weeks in these 10 patients, accrual to this arm would be stopped. Otherwise, 19 additional patients would be accrued for a total of $n = 29$, with an estimated ORR ≥ 30%. This study was designed to have an alpha of 0.0471, and a power of 81%.

The first and last patient were enrolled in May 2015 and January 2017 respectively. As the trial enrolled 34 patients it became clear that unselected patients would not benefit. A decision was made by the study team and accepted by the institutional review board and data safety committee that it would not be in the best interest of the patients to continue this study for lack of efficacy.

**Study treatment, drug administration, safety and response assessment.**
Patients received 400 mg of vorinostat once daily orally for 5 days/week and 20 mg tamoxifen daily in 21-day cycles. Pembrolizumab at 200 mg was given intravenously on day 1, week 1 on cycle 1 (Arm A) or cycle 2 (Arm B) to investigate potential benefit of epigenetic priming to vorinostat until disease progression or unacceptable toxicity. To allow for epigenetic priming, patients initially received vorinostat and tamoxifen for 3 or 9 weeks prior to pembrolizumab dosing. This changed to 3 weeks or no priming due to very rapid progression in the first 5 patients who came off the study before receiving pembrolizumab.

Toxicity was assessed continuously according to the NCI Common Terminology Criteria for Adverse Events Version 4.0 (CTCAE v4.0).

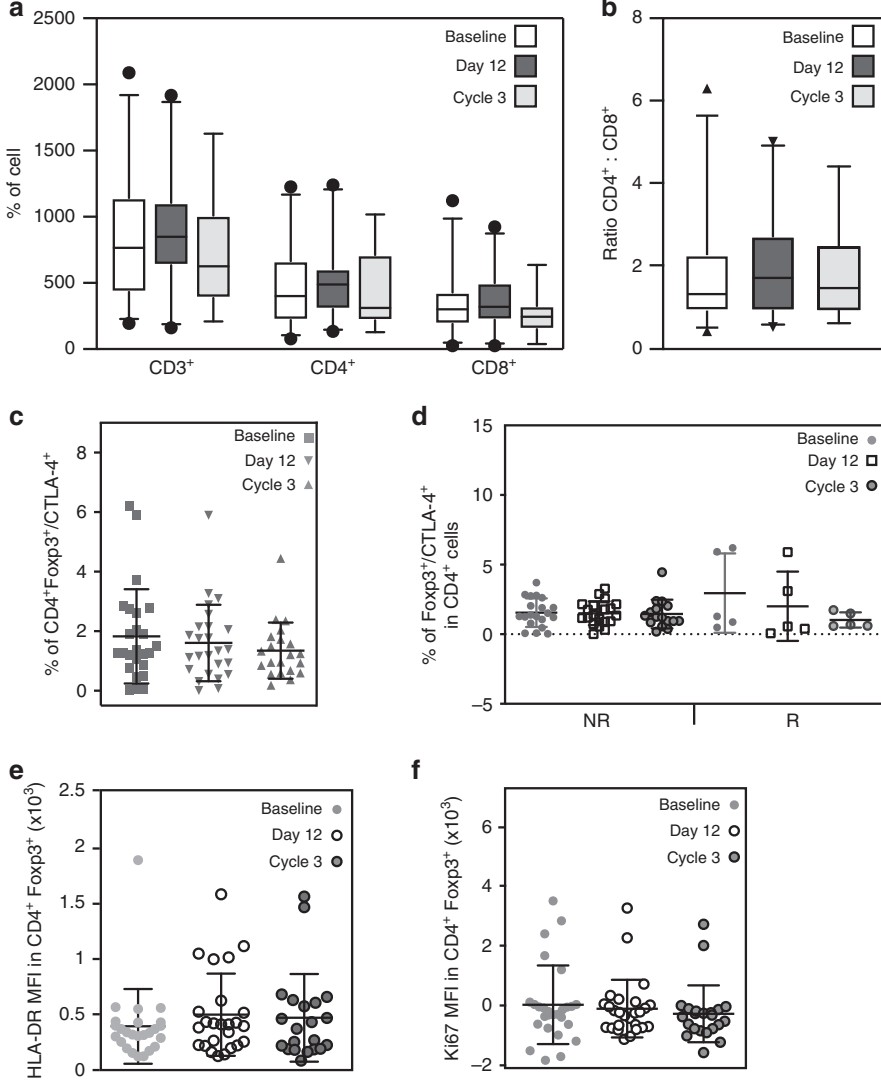

**Fig. 5 Immune cell subsets evaluation in blood was not affected by the treatment.** Quantification of CD3, CD4 and CD8 cells (**a**) and ratio of CD4:CD8 cells (**b**) before beginning of treatment (Baseline), on day 12 of Cycle 1 (Day 12) and at the end of Cycle 3 (Cycle 3) ($n = 25$). The markers represent the 5th percentile, first quartile, median, third quartile, and 95th percentile values, respectively (**a**, **b**). **c** Flow cytometry quantification of Foxp3+CTLA-4+CD4+ Tregs at baseline, on day 12 Cycle 1 (Day 12) and at the end of Cycle 3 for all patients ($n = 25$). **d** Flow cytometry quantification of Foxp3+CTLA-4+ CD4+ Tregs in blood obtained from all the patients with or without a response (R and NR respectively) before treatment (Baseline), on Cycle 1 day 12 and at the end of Cycle 3 ($n = 25$). Flow cytometric quantification of HLA-DR (**e**) and Ki67 (**f**) expression in Foxp3+ CD4+ T-cells at baseline, on day 12 of Cycle 1 and at the end of Cycle 3 ($n = 25$). Data are presented as the mean ± SD. No statistical significance was found in any of the panels, This was determined by one-way ANOVA for **a** (CD3) and **c** and Kruskal-Wallis test for **a** (CD4 and CD8), **b**, **d**–**f**.

**Correlative studies: histone acetylation.** Histone H3 and total lysine acetylation were evaluated in PBMCs obtained at baseline, on day 10(+2) of cycle 1 and at the end of cycle 3 (day 17–19) post-treatment, as a pharmacological marker of HDACi activity. Briefly, cells were spun at 2500 rpm for 10 min, separated from plasma and the pellets were stored at −80 °C until analysis. Once all the samples for one patient were collected, lymphocytes and red blood cells were separated with lysis buffer (BD Pharm Lyse™), following antibody staining to perform flow cytometry analysis. The following antibodies were used: Ghost/Dye v450 viability dye (#13-0863 Tonbo Bioscence); anti-human acetyl histone H3 (Cell signaling cod. #9649); anti-human Acetyl-Lysine (Cell signaling cod. #9441); followed by secondary antibody goat anti-rabbit IgG H + L-AlexaFluor488 (Life Technologies, cod. A11008). Flow cytometry staining protocol was based on a variation of the method of Ronzoni et al.[50]. Briefly, lymphocytes were incubated in Ghost/Dye v450 for 30 min at 4 °C in the dark, then fixed with Paraformaldehyde (PFA) for 15 min at 4 °C in the dark and permeabilized for 5 min at RT in a solution of 0.4% triton X-100 in 0.1% BSA in PBS 1×. Cells were then incubated in primary antibodies, followed by secondary antibody, both for 1 h at RT in the dark. Cells were re-suspended in cold PBS-EDTA 3 mM and kept at 4 °C until analysis. Samples were acquired on BD FACSVerse Flow cytometer with BD FACSuite software. Gates were determined using isotype control antibody staining; live/dead, positive and negative gates were based on II only non-specific staining.

**Correlative studies: immunophenotyping.** Multiparameter flow cytometry was performed on pretreatment (baseline) and post-treatment (cycle 3 day 17–19) fresh biopsies of metastatic sites, when accessible and safe, and on PBMCs (collected on pretreatment, day 10–12 of cycle 1 and day 17–19 of cycle 3).

Freshly isolated samples were minced and digested overnight with buffer consisting of collagenase type IV (Worthington Biochemical Corp., #4188;), DNAse (Sigma-Aldrich, #DN25), 1% penicillin-streptavidin (CCF, #INVZR124), 1% 1 nM Na-pyruvate (Millipore, #TMS-005-C), 1% HEPES (Gibco, #15630-080), 1x non-essential amino acids (Gibco, #11140-050) and 11nM-mercapto-ethanol (Sigma-Aldrich, #M6250) in RPMI medium (Gibco, # 21870092) and collected as single-cell suspensions. PBMCs were suspended in 10% FBS, 1% HEPES, and 1% penicillin-streptavidin in RPMI medium. Approximately $3 \times 10^6$ cells were characterized by a multiparameter flow cytometry analysis performed on a BD LSRFortessa with BD FACSDiva software and analyzed by FlowJo software (BD Biosciences). The flow cytometry immunophenotyping panel was designed based on previous findings[28] and includes the antibodies used, listed in Supplementary Table 1. To standardize voltages over time, Sphero Ultra Rainbow Beads (Spherotech) were used to calibrate and normalize to baseline intensity. Gates were determined using fluorescent minus one (FMO), isotype control antibodies staining and an internal negative control cell population (PD-1 and CTLA-4 expression on CD3- CD4- CD8- cells). Gating strategy is shown in Supplementary Fig. 6.

**PD-L1 staining by immunohistochemistry**. H&E staining and PD-L1 (clone SP263; #790-4905, Roche) protein expression on freshly isolated tumor tissue from pre-treatment and post-cycle 3 biopsies were performed immunohistochemically by the UCSF Tissue Core. A study-dedicated pathologist blindly performed the scoring of PD-L1, evaluating the % of positive cells present in each tissue (Supplementary Fig. 1).

**Efficacy assessment and statistical methods**. The primary endpoints were the evaluation of the safety and tolerability of the combination treatment, the evaluation of the objective response (ORR), defined as complete response (CR) and partial response (PR), and clinical benefits rate [CBR = CR + PR + stable disease (SD) > 6months] by RECIST 1.1 and irRECIST criteria. Secondary endpoints included progression-free survival (PFS), immune response, and overall survival (OS). Correlative endpoints include epigenetic modulation and immune modulation as well as their interaction, together with the evaluation of changes in histone acetylation.

The study employed a 2-arm Simon two-stage design with early stopping rules. All statistical evaluations were performed with Prism software version 8.0 (GraphPad Software, Inc). Data are expressed as mean with standard deviation (±SD) indicated. Data were assessed for normal Gaussian distribution with Shapiro–Wilk test and then analyzed by Mann-Whitney test, Kruskal–Wallis test, Wilcoxon test, ANOVA test or T test, as appropriate. The level of statistical difference was defined as *$p \leq 0.05$; **$p \leq 0.01$; ***$p \leq 0.001$; ****$p \leq 0.0001$; NS = not significant.

**Reporting Summary**. Further information on research design is available in the Nature Research Reporting Summary linked to this article.

## Data availability

All relevant data that support the findings of this study are available from the corresponding author upon reasonable request.

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

## Author contributions

M.T.B. and P.N.M. designed the study, analyzed the data, wrote and edited the manuscript. M.T.B. developed the methodology, performed the experiments and assisted in sample collection. N.P. collected samples and assisted in performing experiments. A.J.C., M.E.M., H.R. and P.N.M. enrolled patients. R.R. and T.D. provided administrative, material and patient support. M.D., A.D. and S.T. assisted in the writing of the manuscript. M.M., S.T., A.D. and M.D.R. contributed to the study design.

## Competing interests

The authors declare no competing interests.
