## [Peer Review File · Nature Communications]

Reviewers' Comments:

Reviewer #1:

Remarks to the Author:

In this interesting work, Terranova-Barberio have shown the importance of CD8+ T cells with an exhausted phenotype in the prediction of response to immunotherapy (pembrolizumab) and also the immunomodulatory effect(s) of HDACi. This is potentially an important observation as it identifies an immune signature other than immune check point expression being a predictor for response to IC inhibitor. It is also one of the few reports that compares concomitant peripheral blood and TILs in terms of numbers and surface markers.

While this work is interesting, and some aspects are novel but there are a few issues which need to be addressed.

General comment: My main concern is the number of cases and the conclusions which are made based on very limited number of cases with borderline significance in some cases. For instance, the conclusion that exhausted CD8+ T cells are higher in responder patients is based on just 3 responder cases. Although I do appreciate that the trial was terminated early and not many responders. I've also found it very difficult and sometime confusing to follow the result section as authors are frequently switching back and forth from clinical findings to pre-clinical studies and to the lab results which makes it challenging to follow and more importantly understand the results, number of samples and methodology.

As mentioned, the concomitant assessment of immune cells in PB and tumour microenvironment is an important aspect of this work, nevertheless, the comparison between the results are very vague and as far as I understand, authors haven't done any correlation study (in statistical terms) between PB and tumour microenvironment's immune signatures.

A few specific comments:

1- The flow cytometry data is based on limited markers and analysis strategy is not clear enough. For instance: How did you define the PD1 expression "percentages" (Page 8, second paragraph)?

Which markers were used to define Tregs? How did you define activated Tregs (page 9), considering that you don't have CD45RA as the classic marker to define Treg subpopulations?

2- I am not convinced that HLADR or ki67 are enough "functional" markers and wondering whether authors have checked the function of Tregs (ie by their cytokine profile of proliferation assay) following exposure to HDACi?

3- The acetylation results are not convincing enough. There are several different immune cells within PBMCs which may have different levels of acetylation with different functional consequences and cannot be detected in total lysine acetylation.

4- Figure 2 a & b, are these results from PB or tumour? It needs to be specified throughout.

Reviewer #2:

Remarks to the Author:

This manuscript describes the results from a phase II clinical trial on the safety and preliminary efficacy of epigenetic immune priming in breast cancer using combined therapy of vorinostat, tamoxifen and pembrolizumab. The trial was halted early due to insufficient efficacy. The manuscript focused on the correlative analysis and explored the effects of co-administration of a HFACi on immune responsive phenotype and potential signatures of response. The study provides some evidence that T-cell exhaustion (CD8+PD-1+/CTLA-4+) and the treatment related depletion of regulatory T-cell (CD4+ Foxp3+/CTLA-4+) may serve as potential signatures for the response of this combination therapy in the study population. The evidence overall is relatively weak given the very small sample size, very small number of patients experienced clinical benefits, possible large numbers of examined potential markers which may result in chance finding and multiple weaknesses in statistical analysis of the data. Specific issues are listed below.

1. Abstract, 2nd paragraph, "... vorinostat-induced depletion of regulatory T-cells..." given the study only examined the combined therapy, it is unclear why it is called vorinostat-induced. Evidence should be provided or remove "vorinostat-induced". Many other places in the results also attributed changes observed to vorinostat which is not appropriate.
2. Page 5, second paragraph, it is said that "... partially exhausted PD-1+/CTLA4+ CD8+ T-cells and their epigenetic modulation were part of preplanned correlative endpoints in blood and tumor samples for this study, as well as comprehensive T-cell immune-profiling", however, partially exhausted PD-1+/CTLA4+ CD8+ T-cells was not a clearly defined correlative endpoint in the study protocol. It is unclear how many potential markers were explored. This information is important as a large number of explored biomarkers could increase the probability of identifying one with statistical significance simply by chance.
3. The study reported five responders, but the reported correlative analysis results were based on analyses of 3 responders, for example, Figures 2D, 3D, 4A, and 4B. It is unclear why there were only 3 responders and how these 3 are different from the other two.
4. The statistical analyses methods were described very vaguely in the stat section. Normality is assumed without actually verifying. Also it is not appropriate to simply say t-test was used given there are different types of t-test and incorrect type could result in erroneous results. Additionally, for some analyses, for example, those related to Figure 4A, 4B, 4C and 4D, and Figure 5 in general, t-tests may not be appropriate. The study team should work with a statistician to ensure the data are analyzed appropriately.
5. Page 5, second paragraph, final sentence, "The percentage of exhausted CTLs was not affected by vorinostat treatment in either responders or non-responders (Figure 3A)", it is unclear how one can reach this conclusion based on figure 3A, where one cannot identify data points of responders and data points of non-responders. Also, line plots would be more appropriate here. Statistical methods used to reach the conclusion should be included. Just saying t-test would not be appropriate.

Reviewer #3:

Remarks to the Author:

Overall, this is a very interesting study. There is limited data to date for activity of immunotherapy in ER+ breast cancer and this study represents a novel treatment approach to add immunotherapy to endocrine therapy with HDACi. The study was stopped early for futility and there were 5 patients of 34 with clinical benefit. The authors demonstrate that T-cell exhaustion was seen in those pts with clinical benefit. Overall interesting, but a small study, with few pts with clinical benefit, and the manuscript needs to be more clearly written.

1. Abstract: Need a line to clearly state trial design and planned primary endpoint; please include antibody used for PDL1 testing

2. Introduction:

a. ORR for KN086 in 1L TNBC with pembrolizumab monotherapy was 21.4% and was 24% in 1L with atezolizumab (Emens L et al, JAMA Onc); would make sure to actually cite the original trials that looked at this and change the range you note to go up to 24%

b. Would state ORR for ER+ PDL1+ was 12%; and can cite negative trial presented at ASCO for Eribulin +/- pembrolizumab in metastatic ER+ disease (Tolaney S et al, ASCO 2019)

c. I think the line should be "vorinostat has been show to restore hormone therapy sensitivity"not resistance

3. Results

a. With regards to prior systemic therapy, would specify this is in metastatic setting, and that it includes hormonal therapy? Would be nice to see prior # chemotherapy for metastatic disease written in text (it is show in Table 1). Also was prior tamoxifen allowed?

b. Please specify how PDL1 testing was done – it states it was done on biopsies—did you have tissue from all patients by biopsy for testing, and what scoring system used? I see SP263 was done and in one section in alludes to staining on tumor cells—did you not look at immune cell scoring

c. What was the study powered to assess – ie. what was the accrual goal and primary objective, and stats for this? Was there an early stopping rule, or how did you decided to stop early? This isn't clear to me in the stats section at the end

d. A CR was seen in 1 of 28 patients who received both vorinostat and pembrolizumab and “was” rather than “were”

i. did this include tamoxifen?

e. “None of the patients having discernible PD-L1 staining in their tumor” -- would change this to state that none of the patients with clinical benefit had PDL1 positivity

4. Correlative Analyses:

a. It would be helpful to provide numbers of samples were looked at.

5. Discussion

a. ...consistent with other studies, showing a very low response rate to PD-L1 checkpoint...

i. Would remove PDL1 as this is seen with PD1 inhibitors as well

ii. And would argue in ER+ breast cancer we don't know that response in PDL1+ tumor would be higher with checkpoint inhibitors given lack of benefit in this subgroup to date, so I would clarify these statements

iii. Overall, the CBR in this study is low, so it is hard to know if further work should be done with this combination even in the subgroup with the exhausted CD8 T-cell signature; it is hard to draw conclusions from just 5 pts with benefit, as pointed out with the limitations

Response to Reviewers:

We would like to thank the reviewers for their kind and thoughtful comments and the recognition that this is a trial with a limited number of patients. While the study is small, the sensitivity and specificity of the tumor samples for the signature is 100%, which was added, upon prompting of the reviewers. The signature in blood has a negative predictive value of 91%, which could make this signature useful for screening patients who have difficult to access tumors, and for study prescreening.

34 patients were enrolled and 28 patients received at least one dose of pembrolizumab. In a group of patients with 5 prior therapies, many had rapid progression and dropped out or died even before they received a single dose of pembrolizumab. The majority of patients had visceral metastases. We hope by addressing all the reviewers' concerns, step by step, this manuscript will be deemed acceptable. However, if further clarification is required or changes requested, we will do our best to address them.

Reviewer 1:	
In this interesting work, Terranova-Barberio have shown the importance of CD8+ T cells with an exhausted phenotype in the prediction of response to immunotherapy (pembrolizumab) and also the immunomodulatory effect(s) of HDACi. This is potentially an important observation as it identifies an immune signature other than immune check point expression being a predictor for response to IC inhibitor. It is also one of the few reports that compares concomitant peripheral blood and TILs in terms of numbers and surface markers. While this work is interesting, and some aspects are novel but there are a few issues which need to be addressed.	
General comment: My main concern is the number of cases and the conclusions which are made based on very limited number of cases with borderline significance in some cases. For instance, the conclusion that exhausted CD8+ T cells are higher in responder patients is based on just 3 responder cases. Although I do appreciate that the trial was terminated early and not many responders. I've also found it very difficult and sometime confusing to follow the result section as authors are frequently switching back and forth from clinical findings to pre-clinical studies and to the lab results which makes it challenging to follow and more importantly understand the results, number of samples and methodology.	The result section was revised incorporating all the specific comments and the general comments, please see detailed areas below.
As mentioned, the concomitant assessment of immune cells in PB and tumour microenvironment is an important aspect of this work, nevertheless, the comparison between the results are very vague and as far as I understand, authors haven't done any correlation study (in statistical terms) between PB and tumour microenvironment's immune signatures.	In response to the reviewers concerns we have added the following: we now performed a Pearson correlation of the relationship between blood and tumors of non-responders, with a significant p-value of 0.0178. None of the evaluable non-responders had a PD1/CTLA4 signature in tumor, rendering the positive and negative predictive value of the test at 100%. The negative predictive value for blood was 91%. We only had three tumor samples of the responders and did not find a correlation in the small sample set. We did not

	initially show any correlation between blood PBMCs and tumor due to the limitative number of tumor specimen available for responders and had included this as limitations in the study in our Discussion section. We have now added the correlation for blood and tumor in Supplementary figure 3 and the positive and negative predictive values in the text.
A few specific comments: 1) The flow cytometry data is based on limited markers and analysis strategy is not clear enough. For instance: How did you define the PD1 expression “percentages” (Page 8, second paragraph)?	The immunophenotyping flow cytometry panel was selected based on our preclinical studies and a similar signature in melanoma. We had previously found in advanced metastatic melanoma patients that the relative abundance of partially exhausted tumor-infiltrating CD8+ T cells predicts response to anti-PD-1 therapy (Daud A et al, J Clin Inv 2016, doi: 10.1172/JCI87324). Thus, our panel included 10 markers (listed in Supplementary Table 1). In response to the reviewer, we have further expanded the method section to clarify the details for gating and identification of partially exhausted CD8 positive T cells (CTLA4+/PD1+) and other populations analyzed (see Supplementary Figure 4, depicting the gating strategy in a tumor specimen of a responder and a non-responder). The word “percentage” defines the percentage of CD8+ cells that were positive for the specific marker PD-1 and CTLA-4. Moreover, we have further detailed the Methods section “Correlative studies: Immunophenotyping” to show gating and definition of PD-1 and CTLA-4 positive frequency in the cells. The text is: “To standardize voltages over time, Sphero Ultra Rainbow Beads (Spherotech) were used to calibrate and normalize to baseline intensity. Gates were determined using fluorescent minus one (FMO), isotype control Abs staining and an internal negative control cell population (PD-1 and CTLA-4 expression on CD3-CD4-CD8- cells). Gating strategy is shown in Supplementary Figure 4.”
Which markers were used to define Tregs? How did you define activated Tregs (page 9), considering that you don’t have CD45RA as the classic marker to define Treg subpopulations?	This is a very relevant point and the definition of Tregs in multiple studies including (Miyara M et al. Immunity, 2009, DOI 10.1016/j.immuni.2009.03.019) is important. In this study, we set out to verify whether our preclinical findings of HDAC related modulation of Foxp3+ CD4 regulatory cells was also seen in patients. We did not define activated Tregs in our study using CD45RA for our immunophenotyping analysis by flow cytometry because it was not part of the panel. We used the same panel for both tumor and blood analysis, which was performed in parallel. CD45RA is

	known not to be present in tissue, as previously described (Sanchez Rodriguez R et al, J Clin Invest 2014, doi: 10.1172/JCI72932). We, therefore, defined Treg as CD4+ FoxP3+ CTLA-4+. We agree with the reviewer that it would have been interesting to have had CD45RA as a marker to investigate this specific aspect in blood. Instead, in the paper, by looking at HLA-DR or Ki67 expression on Tregs, we evaluated their activation or proliferation status, respectively. HLA-DR is broadly recognized as a T cells activation marker and it has been previously found to identify specific subpopulations of activated Treg (Baecher-Allan C et al, J Immunol, 2006, doi: 10.4049/jimmunol.176.8.4622). Ki67 is instead widely used to define proliferation status of multiple kind of cell type. Of note: We analyzed the CD45RA in CD4+ cells in peripheral blood obtained in patients and analyzed by the commercial lab, but saw no differences with treatment.
I am not convinced that HLADR or ki67 are enough “functional” markers and wondering whether authors have checked the function of Tregs (ie by their cytokine profile of proliferation assay) following exposure to HDACi?	We agree with the suggestion that investigating cytokines profile in Treg after HDACi exposure would be very interesting. The correlative studies for the clinical samples were limited and driven by the preclinical findings in our studies in breast cancer. With very limited amount of human tumor tissue, markers were prioritized and did not allowed us to perform more assays. HLA-DR and Ki67 use has been demonstrated in various publications including but not limited to:  - Kestens L, et al. AIDS. 1992, doi: 10.1097/00002030-199208000-00004; - Baecher-Allan C et al, J Immunol, 2006, doi: 10.4049/jimmunol.176.8.4622; - Ferenczi K et al, J Autoimmun, 2000, doi: 10.1006/jaut.1999.0343; - Daud A. et al., J Clin Invest, 2016, doi: 10.1172/JCI87324; - Schacht Revenfeld AL et al, Int J Mol Sci, 2017, doi: 10.3390/ijms18071603; - Miller JD, et al. Immunity, 2008, doi: 10.1016/j.immuni.2008.02.020. Based on this data, HLA-DR and Ki67 were used as functional markers and were felt to be useful in defining the activation or proliferation status of cells in these settings.
3- The acetylation results are not convincing enough. There are several different immune cells within PBMCs, which may have	We don't disagree with the reviewer that histone acetylation is global and different genes may be acetylated differently. Our point here and in previous

different levels of acetylation with different functional consequences and cannot be detected in total lysine acetylation.	studies has been that histone acetylation in patients is a host phenomenon, e.g the ability to acetylate histone is host specific and not determined by drug concentrations, dosing or schedules. The focus of our lab has been to find predictive markers to be able to enrich clinical trials for patients who are more likely to benefit to therapeutic interventions. We have performed both total lysine acetylation and quantified histone H3 acetylation as well. We have previously reviewed this challenge in our editorial (Terranova-Barberio M et al, Oncotarget, 2017, doi: 10.18632/oncotarget.22422). Furthermore we and other have previously published that Histone H3 and H4 acetylation are the most robust tests to identify patients that would likely respond to HDACi therapy. Often, in our previous studies, the ability to sustain histone H3 or H4 acetylation was found to correlate with response, clinical benefit and time to progression (Munster PN et al, British Journal of Cancer, 2009, doi:10.1038/sj.bjc.660529; Munster PN et al, Br J Cancer, 2011, doi: 10.1038/bjc.2011.156; Aggarwal R. et al., J Clin Oncol, 2017, doi: 10.1200/JCO.2016.70.5350); Yardley et al., J Clin Oncol, 2013, doi: 10.1200/JCO.2012.43.7251).
4- Figure 2 a & b, are these results from PB or tumour? It needs to be specified throughout.	Figure 2A and 2B referred to both tumor and blood results and had not been further specified. In Figure 2A we have now added the symbol # next to a patient when a tumor specimen was available to be considered in the analysis. We believe it will not be possible to identify the difference between tumor and blood in Figure 2B, but an identical criterion was used. The Figure legend has been updated too.
Reviewer 2:	
This manuscript describes the results from a phase II clinical trial on the safety and preliminary efficacy of epigenetic immune priming in breast cancer using combined therapy of vorinostat, tamoxifen and pembrolizumab. The trial was halted early due to insufficient efficacy. The manuscript focused on the correlative analysis and explored the effects of co-administration of a HFACi on immune responsive phenotype and potential signatures of response. The study provides some evidence that T-cell exhaustion (CD8+PD-1+/CTLA-4+) and the treatment related depletion of regulatory T-	We thank reviewer 2 for the kind and thoughtful comments and hope our responses will provide the sought answers. We feel it is important to highlight that we did not randomly find this signature of double expression of PD-1 and CTLA-4 in CD8+ T cells as a potential biomarker of response among a broad selection of many markers. The immunophenotyping panel used for our flow cytometry analysis included 10 markers (CD45, CD3, CD4, CD8, Live/Dead discrimination, FoxP3, HLA-DR, Ki67, CTLA-4 and PD-1) of which 5 were phenotypical markers and the remainders were functional markers (excluding L/D), as described in Supplementary Table 1. These 10 markers were used to identify as little as 4

cell (CD4+ Foxp3+/CTLA-4+) may serve as potential signatures for the response of this combination therapy in the study population. The evidence overall is relatively weak given the very small sample size, very small number of patients experienced clinical benefits, possible large numbers of examined potential markers which may result in chance finding and multiple weaknesses in statistical analysis of the data.	subpopulations of immune cells. Thus, we investigated only a very select number of potential markers. Our preclinical data suggested that HDACi would increase CD8 cells and PD-L1 expression in tumors which we did not see and decrease Foxp3+ CD4 cells which we did see. Comparing tumors and PBMCs in a preclinical model, we only found this effect in CD4 cells in the tumor. Furthermore from collaboration with the melanoma group, it was shown that relative abundance of partially exhausted tumor-infiltrating CD8+ T cells predicted response to anti-PD-1 therapy (Daud A et al, J Clin Inv 2016, doi: 10.1172/JCI87324), as specified in our Discussion. This provided the rationale for the correlative studies. We modified the Methods section “Correlative studies: Immunophenotyping” to specify this aspect. However, Daud et al did not find a signature in blood and did not use an HDAC inhibitor to modulate T-regs.
Specific issues are listed below: 1. Abstract, 2nd paragraph, “... vorinostat-induced depletion of regulatory T-cells...” given the study only examined the combined therapy, it is unclear why it is called vorinostat-induced. Evidence should be provided or remove “vorinostat-induced”. Many other places in the results also attributed changes observed to vorinostat which is not appropriate.	We have modified the Abstract and the Results as suggested and renamed it treatment induced.
2. Page 5, second paragraph, it is said that “... partially exhausted PD-1+/CTLA4+ CD8+ T-cells and their epigenetic modulation were part of preplanned correlative endpoints in blood and tumor samples for this study, as well as comprehensive T-cell immune-profiling”, however, partially exhausted PD-1+/CTLA4+ CD8+ T-cells was not a clearly defined correlative endpoint in the study protocol. It is unclear how many potential markers were explored. This information is important as a large number of explored biomarkers could increase the probability of identifying one with statistical significance simply by chance.	We appreciate this concern of the reviewer and we have detailed the number of markers explored as listed in the Supplementary Table 1. The analysis included 10 markers (as specified above in the previous point by reviewer 1). In our protocol one of the Exploratory objectives was “To evaluate inflammatory T cell signature changes in blood and tumor biopsies pre- and post-therapy”. As specified above, we specifically designed our flow cytometry panel based on our clinical and preclinical previous findings (Terranova-Barberio M et al, Oncotarget, 2018, doi: 10.18632/oncotarget.23169; Daud A et al, J Clin Inv 2016, doi: 10.1172/JCI87324).
3. The study reported five responders, but the reported correlative analysis results were based on analyses of 3 responders, for	We have 5 responders. Tumor specimens were accessible in only 3 of the 5 responding patients. Blood samples were available for all five patients.

example, Figures 2D, 3D, 4A, and 4B. It is unclear why there were only 3 responders and how these 3 are different from the other two.	We have modified the Methods section of “Patients selection” that now says “Measurable disease, defined by RECIST 1.1, and pre and post treatment biopsies were required if tumors were accessible and safe for biopsy”.
4. The statistical analyses methods were described very vaguely in the stat section. Normality is assumed without actually verifying. Also it is not appropriate to simply say t-test was used given there are different types of t-test and incorrect type could result in erroneous results. Additionally, for some analyses, for example, those related to Figure 4A, 4B, 4C and 4D, and Figure 5 in general, t-tests may not be appropriate. The study team should work with a statistician to ensure the data are analyzed appropriately.	The study statistician had described the statistics required in the study protocol. In response to reviewer 2, we have added more details to the statistical analysis in the methods and results including tests used and assessment of normality. We have revised the statistic section and figure legend where before not all the information was included.
5. Page 5, second paragraph, final sentence, “The percentage of exhausted CTLs was not affected by vorinostat treatment in either responders or non-responders (Figure 3A)”, it is unclear how one can reach this conclusion based on figure 3A, where one cannot identify data points of responders and data points of non-responders. Also, line plots would be more appropriate here. Statistical methods used to reach the conclusion should be included. Just saying t-test would not be appropriate.	In response we have modified this sentence and modified the figure to show a line plot, where it is possible to distinguish responders (empty square) from not-responders (full circle). Figure legend was modified accordingly.
Reviewer 3:	
Overall, this is a very interesting study. There is limited data to date for activity of immunotherapy in ER+ breast cancer and this study represents a novel treatment approach to add immunotherapy to endocrine therapy with HDACi. The study was stopped early for futility and there were 5 patients of 34 with clinical benefit. The authors demonstrate that T-cell exhaustion was seen in those pts with clinical benefit. Overall interesting, but a small study, with few pts with clinical benefit, and the manuscript needs to be more clearly written.	
1. Abstract: Need a line to clearly state trial design and planned primary endpoint; please include antibody used for PDL1 testing	We have changed the abstract to reflect the suggestions as much as possible in order to not exceed the journal limitations. PD-L1 antibody specifications are now included in the Methods section. Given the space constraint in the abstract we have placed the PD-L1 staining details into the method section and in the Supplementary Figure 1 legend.

2. Introduction: a. ORR for KN086 in 1L TNBC with pembrolizumab monotherapy was 21.4% and was 24% in 1L with atezolizumab (Emens L et al, JAMA Onc); would make sure to actually cite the original trials that looked at this and change the range you note to go up to 24%.	We have updated the ORR % and included the requested citations as references n5.
b. Would state ORR for ER+ PDL1+ was 12%; and can cite negative trial presented at ASCO for Eribulin +/- pembrolizumab in metastatic ER+ disease (Tolaney S et al, ASCO 2019)	We have included the ORR for PD-L1 positive patients suggested and have added the eribulin trial and its citation as suggested as reference n6.
c. I think the line should be “vorinostat has been show to restore hormone therapy sensitivity” not resistance.	We have modified the sentence as suggested.
Results a. With regards to prior systemic therapy, would specify this is in metastatic setting, and that it includes hormonal therapy? Would be nice to see prior # chemotherapy for metastatic disease written in text (it is show in Table 1). Also was prior tamoxifen allowed?	The table 1 has a specific section were we states the number and the type of previous treatments. The table specifies that this is in a metastatic setting. We added in the Methods section “Patient selection” that tamoxifen was included as prior treatment allowed in the enrollment. The text now reads: “Pre and postmenopausal women or men with metastatic ER-positive breast cancer, after progression on at least one hormonal therapy in the metastatic setting and any number of prior chemo- or hormonal therapy, including tamoxifen, were eligible for this trial”. In addition to clarify even further we have modified the Results section “Baseline characteristics and patient disposition” that now reads: “The median number of prior lines of systemic treatment in this metastatic setting was 5 (range 2-13), with 85% of patients having received at least 3 lines of therapy, including hormonal therapy”.
b. Please specify how PDL1 testing was done – it states it was done on biopsies—did you have tissue from all patients by biopsy for testing, and what scoring system used? I see SP263 was done and in one section in alludes to staining on tumor cells—did you not look at immune cell scoring?	PD-L1 test was performed by IHC as described in the Methods section on all the tumor specimens available. Tumor biopsies were mandatory if tissue was safe to access. We have modified the Methods section of “Patients selection” that now says “Measurable disease, defined by RECIST 1.1, and pre and post treatment biopsies were required if tumors were accessible and safe for biopsy“ to clarify this. A pathologist blindly performed the scoring of PD-L1 evaluating the % of positive cells present in each tissue. The IHC was performed on tissues biopsies obtain at the time of each time point (no archival

	tissues) as described in the Methods section “Correlative studies: Immunophenotyping” that reads: Multiparameter flow cytometry was performed on pretreatment (baseline) and post-treatment (cycle 3 day 17-19) fresh biopsies”. To highlight this concept we have modified the Methods section “PD-L1 staining by immunohistochemistry (IHC)” to reflect and include this. PD-L1 staining pre and post vorinostat was performed on tumor cells only, as pre-specified in the protocol. Moreover there was no immune infiltrate in tissue (see Supplementary Figure 1).
c. What was the study powered to assess – ie. what was the accrual goal and primary objective, and stats for this? Was there an early stopping rule, or how did you decided to stop early? This isn’t clear to me in the stats section at the end.	The primary and secondary endpoints are described in the Methods section “Efficacy assessment and Statistical Methods”. The statistics section has been modified to clarify what was performed. We have modified the text (Results section, “Baseline characteristics and patient disposition”) in the manuscript to clarify and highlight the reason we halted the trail. This aspect is stated in our protocol in section 8.1: For each arm: Using an Optimal Simon's two-stage design (Simon, 1989), the null hypothesis (H_0) that the true response rate is 10% will be tested against a one-sided alternative. In the first stage, 10 patients will be accrued and a decision will be made to continue or stop accrual based on the observed objective response rate (ORR). If there are 0 or 1 objective responses at 24 weeks in these 10 patients, accrual to this arm of the study will be stopped. Otherwise, 19 additional patients will be accrued for a total of $n=29$. We estimate that each regimen under evaluation will achieve $ORR \geq 30\%$. This study is designed to have an alpha of 0.0471, and a power of 80.5% to test the null hypothesis H_0: $ORR = 10\%$ against an alternative hypothesis, H_A: $ORR = 30\%$, where ORR is defined as CR, PR, or SD > 24 weeks where arms are enrolled independently. While we passed the first stage in both arms, in a discussion with the breast cancer group, the DSMC and the sponsor we decided that as a group we felt it was unethical to proceed with the trial in an unselected setting.
d. A CR was seen in 1 of patients who received both vorinostat and pembrolizumab and “was” rather than “were”. did this include tamoxifen?	The sentence was changed to: “A complete response was seen in one out of the 27 (3.7%) patients who received all three agents and were evaluable for response”. 28 patients received at least one dose of pembrolimub and 27 were evaluable for response by the definition of the protocol.

e. "None of the patients having discernible PD-L1 staining in their tumor" -- would change this to state that none of the patients with clinical benefit had PDL1 positivity	The sentence has been changed to: "None of the patients with clinical benefit had discernible PD-L1 staining in their tumor."
4. Correlative Analyses: a. It would be helpful to provide numbers of samples were looked at.	All patients had correlative studies in blood. Tumors were available for 23 patients at baseline. The manuscript has been adjusted to reflect the tumor biopsies and blood analysis.
5. Discussion consistent with other studies, showing a very low response rate to PD-L1 checkpoint... i: Would remove PDL1 as this is seen with PD1 inhibitors as well	I: We have modified the text as suggested by the reviewer.
ii. And would argue in ER+ breast cancer we don't know that response in PDL1+ tumor would be higher with checkpoint inhibitors given lack of benefit in this subgroup to date, so I would clarify these statements	li: We have rephrased the text to state that there is uncertainty to the relevance of PD-(L)1 staining in ER positive disease
iii. Overall, the CBR in this study is low, so it is hard to know if further work should be done with this combination even in the subgroup with the exhausted CD8 T-cell signature; it is hard to draw conclusions from just 5 pts with benefit, as pointed out with the limitations.	We could not agree more with the reviewer and this is why the trial was halted early. However, our CR patient had a prolonged CR in her very bulky liver metastases after progression on an Aromatase inhibitor alone, then 2 Aromatase inhibitors and CDK4/6 combinations, and came off the trial for an unrelated stroke. The other long-term responder had been on an Aromatase inhibitor, a SERD and CDK4/6 inhibitor. We feel that in a preselect group of patients, immune checkpoint inhibitors and HDAC inhibitors may have a path forward, which will be explored in a randomized trial with preselection. This has been more clearly stated in the discussion.

Reviewers' Comments:

Reviewer #1:

Remarks to the Author:

I have read the latest version of the manuscript and authors' responses to comments. Some of the questions and queries are now adequately addressed by authors and tumour types are better marked in figures. Their response to the question regarding the acetylation is also convincing. While I may somehow disagree with their definition of Treg subset but their argument is scientific and it's mainly a difference of opinion which is fine. The main remaining issue is the number of patients and the conclusions based on the small number of cases which cannot be addressed as the trial is now closed.

Reviewer #2:

Remarks to the Author:

The authors have adequately addressed my concerns.

Reviewer #3:

Remarks to the Author:

I appreciate the attention to addressing all of the reviewers comments in details and updated the manuscript accordingly.